



# 1 A revised mineral dust emission scheme in GEOS-Chem: 2 improvements in dust simulations over China

Rong Tian[1], Xiaoyan Ma[1*], Jianqi Zhao[1]
[1]Collaborative Innovation Center on Forecast and Evaluation of Meteorological Disasters (CIC-FEMD)/Key Laboratory
for Aerosol-Cloud-Precipitation of China Meteorological Administration, Nanjing University of Information Science &
Technology, Nanjing 210044, China
*Correspondence to*: Xiaoyan Ma (xma@nuist.edu.cn)
**Abstract:**
Mineral dust plays a significant role in climate change and air quality, but large uncertainties remain in terms of dust
emission prediction. In this study, we improved the treatments of dust emission process in a Global 3-D Chemical Transport
model (GEOS-Chem) v12.6.0, by incorporating the geographical variation of aerodynamic roughness length ($Z_0$), smooth
roughness length ($Z_{0s}$), soil texture, introducing Owen effect and Lu and Shao (1999) formulation of sandblasting efficiency
α. To investigate the impact of the modifications incorporated in the model, several sensitivity simulations were performed
for a severe dust storm during March 27, 2015 to April 2, 2015 over northern China. Results show that simulated threshold
friction velocity is very sensitive to the updated $Z_0$ and $Z_{0s}$ field, with the relative difference ranging from 10% to 60%
compared to the original model with uniform value. An inclusion of Owen effect leads to an increase in surface friction
velocity, which mainly occurs in the arid and semi-arid regions of northwest China. The substitution of fixed value of α
assumed in original scheme with one varying with friction velocity and soil texture based on observations reduces α by 50%
on average, especially over regions with sand texture. Comparisons of sensitivity simulations and measurements show that
the revised scheme with the implement of updates provides more realistic threshold friction velocities and $PM_{10}$ mass
concentrations. The performance of the improved model has been evaluated against surface $PM_{10}$ observations as well as
MODIS aerosol optical depth (AOD) values, showing that the spatial and temporal variation of mineral dust are better
captured by the revised scheme. Due to the inclusion of the improvement, average $PM_{10}$ concentrations at observational
sites are more comparable to the observations, and the average mean bias (MB) and normalized mean bias (NMB) values
are reduced from -196.29μg $m^{-3}$ and -52.79% to -47.72μg $m^{-3}$ and -22.46% respectively. Our study suggests that the
erodibility factor, sandblasting efficiency and soil-related properties which are simply assumed in the empirical scheme
may lack physical mechanism and spatial-temporal representative. Further study and measurements should be conducted
to obtain more realistic and detailed map of these parameters in order to improve dust representation in the model.

## 31 1 Introduction

Mineral dust is typically produced by wind erosion from regions with arid and semi-arid surfaces in the world and
exerts significant impacts on the atmospheric radiation balance (Tegen et al., 1996; DeMott et al., 2010; Kumar et al., 2014;
Saidou Chaibou et al., 2020a), climate (DeMott et al., 2003; Mahowald and Kiehl, 2003; Zhao et al., 2012; Chen et al.,
2014; Chin et al., 2014), air quality (Giannadaki et al., 2014; Tian et al., 2019) and human health (Goudie, 2014; Tong et
al., 2017). Dust emission process has been recognized as a leading contributor to dust aerosol loading. Global mineral dust
particles are mainly emitted from North Africa, the Arabian Peninsula, Central Asia, East Asia, Australia and North





America, with East Asia (including the Gobi and Taklimakan deserts) accounting for ~20% of the global dust emission
(Ginoux et al., 2004; Nagashima et al., 2016).
In order to properly reproduce dust emission process, many dust emission schemes have been developed and
implemented in both global and regional chemical transport models (CTMs) (e.g., Marticorena and Bergametti, 1995; Lu
and Shao, 1999; Alfaro and Gomes, 2001; Shao, 2001, 2004; Shao et al., 2011; Zender et al., 2003; Kok, 2011a, 2011b).
Nevertheless, some intercomparison studies demonstrated that there are large discrepancies among different dust emission
models (Uno et al., 2006; Todd et al., 2008; Huneeus et al., 2011; Su and Fung, 2015; Ridley et al., 2016; Chen et al., 2017;
Chen et al., 2019; Ma et al., 2019; Wu et al., 2019; Saidou Chaibou et al., 2020b; Zhao et al., 2020). Ma et al. (2019)
quantitatively evaluated the performance of three dust schemes in WRF-Chem, two schemes in both CHIMERE and
CMAQ, and one scheme in CAMx during a dust episode over northern China. Large differences between observed surface
$PM_{10}$ concentrations and modelling results of each model were found. Among schemes in WRF-Chem, AFWA and
UOC_Shao2004 are better correlated with observations compared to GOCART but tend to overestimate dust
concentrations. Kang et al. (2011) compared the performance of three dust emission schemes in WRF-Chem over East
Asia, showing that the difference of dust emission fluxes between three schemes ranges from an order of $10^1$ to $10^2$. Ridley
et al. (2016) showed that the estimated global dust AOD vary by over a factor of 5 among four global models (including
GEOS-Chem, WRF-Chem, CESM and MERRAero), and dust emissions across North Africa are overestimated while
emissions from Asia and the Middle East are underestimated overall. An intercomparison of 14 CTMs as part of the Model
Inter-Comparison Study for Asia (MICS-Asia) phase III project (Chen et al., 2019) showed that nearly all participant
models underestimate $PM_{10}$ levels and current CTMs have difficulty producing similar dust emissions when adopting
different dust schemes.
The uncertainties in dust emission models can be attributed to a number of issues, such as threshold friction velocity,
surface wind speed, soil texture, particle size distribution, other soil/surface parameters (e.g., soil moisture, vegetation
cover, aerodynamic roughness length) and different physical mechanisms (Tegen, 2003; Zhao et al., 2013; Liu et al., 2018;
Chen et al., 2019). Darmenova et al. (2009) conducted a detailed comparison between two schemes developed by
Marticorena and Bergametti (1995) and Shao et al. (1996), indicating that wind friction velocity is a significant factor in
simulating dust emission while the aerodynamic roughness length as well as vegetation cover may play an important role
at higher wind speed. Many sensitivity experiments have been conducted and shown that the modeled threshold friction
velocity can be modified by soil moisture (Cheng et al., 2008; Mokhtari et al., 2012; Gherboudj et al., 2015; Ju et al., 2018),
soil texture (Menut et al., 2013; Gherboudj et al., 2015; Perlwitz et al., 2015a, 2015b; Kontos et al., 2018) and surface
roughness (Cheng et al., 2008; Astitha et al., 2012; Menut et al., 2013), which in turn affects the predicted dust emission.
In addition, a more accurate value of sandblasting mass efficiency (α) has been reported to be a crucial factor for a better
performance of dust emission flux (Mokhtari et al., 2012; Klingmüller et al., 2018; Kontos et al., 2018; Ma et al., 2019).
Based on the above studies, it is necessary to take these key parameters, including soil-related properties and empirical
input parameters, into fully consideration in a dust emission parameterization. Unfortunately, due to limited observations,
many of these parameters are not well included in the model. For example, most dust models simply assume a constant
values of aerodynamic roughness length and soil clay fraction (Ginoux et al., 2001; Tegen et al., 2002; Zender et al., 2003),
ignoring the temporal and spatial variability of them, which may cause uncertainties to the estimated surface friction
velocity and threshold friction velocity. During recent decades, with the development of observation technology, the
detailed information on the surface characteristics appropriate for global and regional models have been provided (Laurent
et al., 2005, 2008; Prigent et al., 2005, 2012; Shangguan et al., 2014; Perlwitz et al., 2015a, 2015b). Therefore, adopting
more accurate and detailed soil datasets is expected to improve the dust model performance.
In this study, we present an improvement of the dust emission scheme in GEOS-Chem model by incorporating the
updated soil texture and aerodynamic roughness length with spatial variability, Owen effect, drag partition correction factor
as well as the updated formulation of sandblasting efficiency, which together significantly improve the prediction of dust



emission flux and concentrations over China. The objective is to obtain more realistic surface friction velocity ($u_*$) and
threshold friction velocity ($u_{*t}$) by considering the effect of the soil moisture, surface roughness and soil texture, thus
improving the representation of dust emission in the model.
Section 2 gives a detailed description of GEOS-Chem model and the modifications of the improved scheme as well
as numerical experiments and data description. Sensitivity results are compared in Section 3.1 to examine the impacts of
the modifications. Section 3.2 presents the comparisons of the improved scheme and original version with observations, to
evaluate the performance of the improved scheme. Uncertainties, limitations, and future improvements of the emission
scheme are discussed in Section 3.3, followed by a summary in Section 4.
**2 Model and measurements**
**2.1 Model description**
The GEOS-Chem model is a global three-dimensional chemical transport model driven by assimilated meteorology.
In this work, we use v12.6.0 of GEOS-Chem driven by GEOS-FP assimilated meteorological field with a spatial resolution
of $0.25° \times 0.3125°$ and 72 vertical levels for China region (15-55°N, 70-140°E) during the period of March 27, 2015 to
April 2, 2015. The lateral boundary conditions is provided by a global simulation of $2° \times 2.5°$.
GEOS-Chem includes detailed atmospheric chemical mechanism and online aerosol calculations. In this work, we
simulate the dust emission with a combination of the dust entrainment and deposition (DEAD) mobilization scheme
(Zender et al., 2003) and Global Ozone Chemistry Aerosol Radiation and Transport (GOCART) source function. Dry
deposition velocities for dust aerosols are computed with the gravitational settling scheme of Fairlie et al. (2007) and
aerosol deposition scheme from Zhang et al. (2001). Wet deposition scheme, which includes scavenging in convective
updrafts, as well as rainout and washout of soluble tracers, is described in Liu et al. (2001). Aerosol optical depth is derived
online from aerosol concentrations with externally-mixed assumption using RH-dependent aerosol optical properties from
Latimer and Martin (2019). Dust optics are from Ridley et al. (2012).
**2.2 Improvement on the dust emission scheme in GEOS-Chem**
The standard dust emission scheme in GEOS-Chem is based on a semi-empirical formulation developed by Zender et
al. (2003) and is combined with GOCART source function (Ginoux et al., 2001). In this scheme, the vertical dust flux ($F$)
is proportional to the horizontal saltation flux ($Q_s$), which is the function of surface friction velocity ($u_*$) and threshold
friction velocity ($u_{*t}$):

$$F = (1 - A_s)S\alpha Q_s \tag{1}$$

$$Q_s = C_z \frac{\rho_{air}}{g} u_*^3 (1 - \frac{u_{*t}}{u_*})(1 + \frac{u_{*t}}{u_*})^2 \quad u_* > u_{*t} \tag{2}$$

where $\alpha$ is the vertical-to-horizontal flux ratio or sandblasting efficiency, based on the soil clay content (Marticorena and
Bergametti, 1995). S is based on GOCART source function (see Fig. S1), also named as the soil erodibility factor,
representing the grid cell fraction of the bare land suitable for mobilization. $A_s$ is the fraction of snow-covered surface. $C_z$
is the saltation constant ($C_z$=2.61).
According to the equation, $u_{*t}$, $u_*$ as well as $\alpha$ are the key input parameters in the accurate prediction of dust
emission flux. $u_{*t}$ is used to describe the characteristics of soil and land surface condition, representing the resistance of
surface to wind erosion. In the standard dust scheme, $u_{*t}$ is calculated using a semi-empirical formulation as a function
of air density and soil particle density (Iversen and White, 1982). Furthermore, two correction terms, including soil
moisture correction (Fécan et al., 1999) and drag partition correction (Marticorena and Bergametti, 1995), are also applied
to modify $u_{*t}$. It should be noted that in the original scheme, the drag partition correction term is eliminated.
$u_*$ is a description of surface wind speed, which mainly depends on 10m wind speed taken from meteorological field





assuming neutral stability (Bonan, 1996). Owen effect, which represents a positive feedback between saltation process and
friction speed (Owen, 1964), is often adopted in models to modify $u_*$. However, Owen effect is eliminated in the original
scheme.
Sandblasting efficiency α is parameterized according to the empirical relation described by Marticorena and
Bergametti (1995) (MB95), which depends on the soil clay content ($M_{clay}$) and is restricted to $M_{clay} < 20\%$:
$$\alpha = 100e^{(134M_{clay}-6)ln10} \tag{3}$$
However, in the global model, α tends to be overly sensitive to $M_{clay}$. Due to this reason, a globally fixed value of
$M_{clay} = 20\%$ is assumed in current model (Zender et al., 2003).
It should be noted that, some input parameters, data or formulations are quite simplified and need to be improved
based on the original dust scheme described above. For example, the aerodynamic roughness length ($Z_0$), the smooth
roughness length ($Z_{0s}$) as well as the mass fraction of clay in the soil ($M_{clay}$) are assumed as a constant uniformly, despite
the fact that it may vary with time and location. As a result, the simulation of related processes, such as drag partition effect
or soil moisture effect, may lack spatial representation. Therefore further modifications on these variables should be made
in order to obtain more realistic dust emission. Fig. S2 presents the schematic diagram of the dust emission schemes in the
standard model and the modifications incorporated in this study. The details of the parameterization options and required
input parameters are presented in following sections.
**2.2.1 Soil Type and Soil Texture Data**
In the model, $M_{clay}$ can have an impact on $u_{*t}$ through modifying soil moisture correction term, thus influencing the
modeled dust emission flux. The soil moisture correction term, defined as $f_w$, is parameterized according to Fécan et al.
(1999), which accounts for the increase of $u_{*t}$ with soil moisture content.
$$f_w = \begin{cases} 1 & w < w' \\ [1 + 1.21(w - w')^{0.68}] & w \geq w' \end{cases} \tag{4}$$
$$w'(\%) = a\left(0.0014M_{clay}^2 + 0.17M_{clay}\right) \tag{5}$$
where $w$ is gravimetric soil moisture and $w'$ is soil residual moisture.
With the increase of soil moisture, soil cohesion can be enhanced, particularly over regions with high clay content,
thus inhibiting sandblasting process to some extent. However, as stated above, $M_{clay}$ is assumed as a constant equal to 20%
in the original scheme, which can cause uncertainty in dust prediction. In the improved scheme, we employ the gridded
data of clay content from the Global Soil Dataset for use in Earth System Models (GSDE) (Shangguan et al., 2014), which
is based on the Soil Map of the World and various regional and national soil databases. Fig. 1 shows the updated $M_{clay}$ from
Shangguan et al. (2014) with the horizontal resolution of $2° \times 2.5°$ at the global scale. Compared to the original fixed value
of 20%, the updated $M_{clay}$ is generally lower in most of the dust source areas over East Asia.
**2.2.2 Surface roughness length**
The drag partition is used to describe the impact of roughness elements (such as rocks, pebbles, vegetation, etc.) on
$u_{*t}$. According to Marticorena and Bergametti (1995), the roughness correction term, $f_d$, is a function of the aerodynamic
roughness length $Z_0$ and the smooth roughness length ($Z_{0s}$):
$$f_d = 1 - \frac{ln(\frac{z_0}{z_{0s}})}{ln\left[0.7\left(\frac{12255cm}{z_{0s}}\right)^{0.8}\right]} \tag{6}$$
where the required roughness lengths are set as the constant values of $Z_0 = 0.01$ cm and $Z_{0s} = 0.0033$ cm globally.
$Z_0$ represents the roughness length of the overlying non-erodible elements (solid obstacles, such as rocks), which
transfers part of the wind momentum from the atmosphere to the surface, dissipating the shear force for particle saltation.
Prigent et al. (2005) derived global aerodynamic roughness length in arid and semi-arid areas which are retrieved from the
ERS-1 satellite measurements with a horizontal resolution of $0.25° \times 0.25°$. Here we apply the global monthly mean $Z_0$





fields provided by Prigent et al. (2005) and then re-grid the map to $2° \times 2.5°$ horizontal resolution for the incorporation
into GEOS-Chem. As Fig. 2 shows, compared to the fixed constant assumed in the original version, the updated global $Z_0$
is generally higher. Over northern China, the $Z_0$ value ranges from approximately 0.01cm over desert regions to 0.07cm.
$Z_{0s}$ characterizes the roughness length of the uncovered, bare erodible surface. Instead of setting $Z_{0s}$ to a fixed value,
some studies suggested that $Z_{0s}$ can be estimated as 1/30 of the coarse-mode mass median diameter (MMD) of soil particles,
which will provide more realistic representation of soil texture distribution (Marticorena and Bergametti, 1995; Laurent et
al., 2006; Mokhtari et al., 2012). In the improved version, we adopt this empirical formula, based on updated soil texture
classification (Mokhtari et al., 2012; Xi et al., 2015), to estimate $Z_{0s}$:
$$z_{0s} = MMD/30 \tag{7}$$

where MMD is the median diameter of the coarsest mode for various soil textures shown in Table 1. The corresponding
$Z_{0s}$ for different soil types are listed in Table 1 and its global distribution is shown in Fig. 3. The soil texture map is obtained
based on the Harmonized World Soil Database (HWSD; http://www.iiasa.ac.at/Research/LUC/External-World-soil-
database/HTML), which provides global sand, silt, and clay contents at 30 arc-second resolution. The soil texture dataset
is re-gridded to $2° \times 2.5°$ resolution, and then is applied to identify the global soil texture by using the United States
Department of Agriculture (USDA) soil texture triangle (based on the amount of sand, clay, and silt contents;
http://soils.usda.gov/technical/aids/investigations/texture/). There are 12 classes of soil defined by USDA. It can be seen
from Fig. 4 that loam, sandy loam and clay loam, are the dominant soil types over China. Among them, sandy loam and
loam occupy the major part of northwest China.
**2.2.3 Sandblasting efficiency α**
Sandblasting efficiency α is important in the dust emission calculation as it is used to convert the horizontal saltation
mass flux to a vertical dust mass flux. In the original scheme, α is simply expressed as a function of $M_{clay}$, which is a fixed
value of 20%. The assumption in the original scheme might cause uncertainty in modeled flux and make the spatial
variation less representative (Mokhtari et al., 2012).
In order to reduce this uncertainty, a more physically-based function from Lu and Shao (1999) (LS99) is adopted in
our study. Based on wind tunnel experiments carried out by Rice et al. (1996a, b), Lu and Shao (1999) derived the
expression of α through theoretical calculation and some simplifications:
$$\alpha = \frac{C_\alpha g f \rho_b}{2p}(0.24 + C_\beta u_* \sqrt{\frac{\rho_p}{p}}) \tag{8}$$

where $f$ is the fine particles content in the soil volume, $p$ is soil plastic pressure, which represents the magnitude of the
surface resistance (N m$^{-2}$), $\rho_b$ and $\rho_p$ are the bulk soil density and particle density, respectively, $g$ is the gravitational
acceleration (m s$^{-2}$), u∗ is friction velocity (m s$^{-1}$), and $C_\alpha$ and $C_\beta$ are empirical constants. Among these parameters, the
values of $\rho_b$ and $p$ depend upon different soil textures. Some studies (Kang et al., 2011; Foroutan et al., 2017; Ma et al.,
2019) have implemented this formulation in the model and proposed the proper range of these parameters over different
soil types.
Many measurements from laboratory experiments and field observations have demonstrated the close relationship
between α and $u_*$ (Gillette et al., 1997; Gomes et al., 2003; Rajot et al., 2003; Roney and White, 2006; Macpherson et al.,
2008; Panebianco et al., 2016; Zhang et al., 2016). To improve the original scheme, we extract α from these measurements
over different soil types, based on the expression of LS99, as depicted in Fig. 5 and Table 2.
**2.3 Experiments design**
Several sensitivity experiments (Table 3) are conducted to assess the model performance of the modifications in the
improved scheme. Control is the control run with the dust emission scheme originally implemented by Fairlie et al. (2007).
Sen_mclay, Sen_owen, Sen_ratio, Sen_drag and Sen_Z₀Z₀s are the same as the control run but including the modification





of $M_{clay}$, Owen effect, sandblasting efficiency, drag partition effect and updated surface roughness length ($Z_0$ and $Z_{0s}$)
respectively. Sen_all represents the simulation with the improved scheme which accounts for all the modification described
above.

**2.4 Measurements**

The data used in this study includes the Moderate Resolution Imaging Spectrometer (MODIS) Level 3 AOD data,
hourly observational data of surface $PM_{10}$ concentration, and meteorological field taken from the Meteorological
Information Comprehensive Analysis and Process System (MICAPS). The data used in this study is for the period of March
27, 2015 to April 2, 2015.
MODIS aerosol products are used to evaluate model results of AOD. MODIS AOD at 550 nm is obtained from the
daily level-3 product from Aqua satellites (MYD08_D3, 1°×1° gridded data) and is combined with Deep Blue retrievals
which can provide AOD over bright surfaces (i.e., desert regions).
Hourly surface observed $PM_{10}$ concentration data, collected from about 1000 environmental monitoring stations
maintained by Chinese Ministry of Environmental Protection (MEP; http://datacenter.mep.gov.cn), is used to validate the
model performance of surface dust concentrations.
Meteorological fields of wind speed taken from the Meteorological Information Combine Analysis and Process system
(Micaps) developed by the Chinese National Meteorological Center (NMC) are used for evaluation of wind field in the
model. Fig. S3-S4 show that the 10m wind field used in the model scheme generally agree well with the Micaps
observations over most sites. However, comparisons of averaged surface wind field between the model input and
observations (Fig. S5) show that although the circulation patterns in the model are identical with the observations, surface
wind speed in the model tend to be larger than observations, especially over western and northeastern Inner Mongolia.

**3 Results and discussion**

**3.1 Sensitivity study**

In order to assess the sensitivity of the dust emission to the modified input parameters or physical processes, several
numerical experiments are conducted and compared. Fig. 6a presents the relative difference (%) of averaged $u_{*t}$ during
study period between these sensitivity simulations and the control run. The $u_{*t}$ simulated by Control run are generally
small, with values less than 0.3m/s (not shown). Wu et al. (2013) indicated that $u_{*t}$ over source regions in northern China
calculated by Zender et al. (2003) are generally lower (with values ranging from 0.2 to 0.25 m/s) than the measurement
(with values ranging from 0.34 to 0.69 m/s) and the values calculated by Shao (2004), which is closer to the observations.
The sensitivity simulations show that the update of $M_{clay}$ in Sen_mclay can lead to higher $u_{*t}$ over northern China and
lower $u_{*t}$ over southern China than the control simulation, which overestimates $M_{clay}$ over northern China while
underestimates it over southern China by setting $M_{clay}$ to 20%. In northern China, particularly in arid and semi-arid regions,
the updated $M_{clay}$ will decrease the soil moisture threshold $w'$ and increase soil moisture term $f_w$, thus leading to a slight
increase in $u_{*t}$ (with magnitude <10%). The inhibition of dust emission by surface roughness elements is not taken into
account in the original scheme, i.e., $f_d=1$. Some studies (Darmenova et al., 2009; Menut et al., 2013) have demonstrated
$f_d$ as a function of $Z_0$ and $Z_{0s}$, implying that $f_d$ increase with $Z_0$ and decrease with $Z_{0s}$. Compared to the fixed values used
in the original scheme, updated $Z_0$ field used in Sen_ $Z_0Z_{0s}$ are generally larger and updated $Z_{0s}$ field are smaller. Therefore,
$f_d$ are increased significantly, particularly over the regions with non-erodible elements (larger $Z_0$). Result shows that $u_{*t}$
is increased when considering the drag partition effect (increased by 10% in Sen_drag with constant $Z_0$ field), particularly
with the updated $Z_0$ and $Z_{0s}$ field (increased by 10%~60% in Sen_ $Z_0Z_{0s}$). In general, due to the inclusion of $Z_0$, $Z_{0s}$ and
$M_{clay}$, $f_d$ and $f_w$ are modified, which results in significant alteration in $u_{*t}$ (ranging from -8%~72% in Sen_all) over
China. It can be found that the modification of $f_d$ due to updated $Z_0$ and $Z_{0s}$ makes more contribution to the increase in





$u_{*t}$.
Relative difference of $u_*$ with respect to the control run are also compared in Fig. 6b. Considering Owen effect in
Sen_owen leads to an increase in $u_*$ by 0%~39%, especially over northwest China where surface wind is strong. Modeled
$u_*$ is obtained from u10m and $Z_0$ under neutral conditions (Bonan, 1996). It can be seen that updated $Z_0$ in Sen_ $Z_0Z_{0s}$ can
modify $u_*$ by influencing the boundary-layer exchange properties. $u_*$ over northern China is generally increased by
10%~22% with higher values of $Z_0$ in Sen_ $Z_0Z_{0s}$, while it is slightly decreased over Taklimakan and Gobi deserts. In
Sen_all, modeled $u_*$ is increased by 5%~50% over most parts of China.
Fig. 6c presents the percentage difference in terms of sandblasting efficiency $\alpha$. In the original version, $\alpha$ is set as a
uniformly constant value (around 0.04) due to the assumption of a fixed $M_{clay}$. In Sen_ratio and Sen_all, $u_*$-dependent-
ratio following LS99, which varies with different soil texture according to observations, is adopted. On average, $\alpha$ is
decreased by 50% with the modification in Sen_ratio and Sen_all. The largest reduction occur over regions with sand
texture such as over Taklimakan and Gobi Desert.
As seen from Fig. 6d, the simulated dust emission flux ($F$) vary significantly among different experiments. Due to the
inclusion of updated $M_{clay}$, soil moisture term increases in Sen_mclay, which leads to higher $u_{*t}$ and lower $F$ over most
regions. Accounting for Owen effect in Sen_owen results in an increase in $F$ of 0%~314%, particularly over northern part
of Gansu Province and northwestern Inner Mongolia. A significant reduction in arid and semi-arid regions of northern
China is caused by updated $\alpha$ (Sen_ratio). In Sen_drag and Sen_ $Z_0Z_{0s}$, $F$ are influenced by -100% ~-4% and -100%~50%
respectively as a result of the inclusion of $f_d$ with constant $Z_0$ and updated $Z_0$, $Z_{0s}$ respectively. Due to the combined
effects of the modifications, $F$ simulated by Sen_all is generally reduced over northern China, except
in some regions of northwest China, where Owen effect plays a dominant role.
Four sites closer to dust source area or significantly influenced by dust-storms (Beijing, Huhehaote, Jiuquan and Kuele,
locations shown in Fig. S1) are selected to evaluate the performance of control and sensitivity simulations. Comparisons
of the modeled $u_{*t}$ (Fig. 7) show that in all sites, modeled $u_{*t}$ are increased in Sen_mclay, Sen_drag, Sen_ $Z_0Z_{0s}$ and
Sen_all, compared with the original model, with the highest $u_{*t}$ simulated by Sen_all. Modeled $u_{*t}$ increase from
0.22~0.25m/s in Control to 0.32~0.37m/s in Sen_all. The reported $u_{*t}$ values over arid and semi-arid regions of China
are around 0.3~0.5m/s (Wang et al., 2009). Wu et al. (2013) summarized that $u_{*t}$ range from 0.34~0.69m/s over East Asia
and indicated that $u_{*t}$ calculated by Zender et al. (2003) are relatively lower, ranging from 0.2m/s to 0.25m/s. It is apparent
that modeled $u_{*t}$ are greatly increased in the revised simulation, which is much closer to the observed values. This
improvement is mainly attributed to the update of $Z_0$ and $Z_{0s}$. Comparisons between the modeled averaged $PM_{10}$
concentrations and the observational values in four sites show that $PM_{10}$ levels simulated by Sen_all are closer to
observations than many other cases. In summary, Sen_all shows the better agreement with the observations in terms of
$u_{*t}$ and $PM_{10}$ concentrations.
**3.2 Comparison between the improved scheme and the original scheme with observations**
In order to validate the model performance of the improved scheme, time series of the observed surface $PM_{10}$
concentrations are compared against the modeled values from Control (the original scheme) and Sen_all (the improved
version) during a dust episode from 27 March to 2 April of 2015 over northern China. The intensity and evolution of this
dust event has been described by Wang et al. (2017), illustrating that dust particles were mainly emitted from Mongolia
and Inner Mongolia province of China and a dust backflow event took place over North China on March 29. Fig. 8 compares
the hourly modeled $PM_{10}$ concentrations and observed values for nine selected sites (locations shown in Fig. S4), which
are closer to the dust sources or severely affected by the dust event. It shows that the dust concentrations are generally
underestimated in Control run, particularly when dust concentrations are quite high, indicating that the original scheme has
difficulty in accurately reproducing the dust emission process. Sen_all generally reproduce the $PM_{10}$ levels better than
Control run. Both the magnitude and the temporal evolution of $PM_{10}$ concentrations are captured in Sen_all quite well,





with peak values much closer to the observations. Among these sites, Sen_all shows better performance over North China,
e.g., Beijing, Tianjin and Huhehaote. But both Control run and Sen_all fail to capture the peak values from 29 March to 30
March. During this period, dust particles, mixed with anthropogenic pollutants, flew back due to the south wind over North
China (Wang et al., 2017). Uncertainties in the meteorological field and dust heterogeneous reactions in the model may
cause the model bias.
For specific periods, however, modeled peak values of some sites occur earlier (several hours) than the observations
at some sites (e.g., Beijing and Tianjin in 28 March), which could be considered as a result from the uncertainty in the wind
field used in the model. It shows that the surface wind is stronger in the model than the observations (Fig. S3), which may
lead to stronger transport of the dust from source regions to downwind areas such as Beijing, Tianjin and Kuele. Instead,
modeled and observed peak values of some sites in the source regions (e.g., Huhehaote, Xilinguole and Hami) almost
simultaneously occur.
In order to quantify the performance of the model result, some statistical parameters, including the mean values,
correlation coefficient (R), mean bias (MB), normalized mean bias (NMB), are calculated and listed in this paper. The
statistical performance for the modeled surface $PM_{10}$ concentrations from Control run and Sen_all against observations
are presented in Table 4. It shows that dust concentrations at all selected sites are significantly underestimated in Control
run, especially over northwest regions, with the MB and NMB values ranging from -163.5μg $m^{-3}$ to -503.61μg $m^{-3}$ and
-64.61% to -68.48% respectively. It is obvious that Sen_all with updated modification greatly improves the dust
concentration prediction, with mean values more comparable to the observations, and the average MB and NMB values
reduce from -196.29 μg $m^{-3}$ and -52.79% in Control run to -47.72μg $m^{-3}$ and -22.46% respectively. The largest
improvement occurs at northwest stations (e.g. Hami, Akesu and Kuele), which are located close to Taklimakan desert.
Over other regions, such as North China (e.g., Beijing, Tianjin, Huhehaote and Xilinguole), the model performance of
Sen_all are slightly better than Control run.
Although the MB and NMB values of most stations are generally lower and the mean values are much closer to
observation for Sen_all simulation, i.e., modifications included improve the underestimation in Control run to some extent,
the dust concentrations are still generally underestimated. For stations closer to Gobi desert, such as Xilinguole, Jiuquan
and Baiyin, dust concentrations are greatly underestimated with NMB<-30%, which is likely due to the uncertainty in the
erodibility factor over Gobi desert used in our study (Ginoux et al., 2001). Similarly, Su and Fung (2015) evaluated the
performance of dust emission schemes in WRF-Chem over East Asia, pointing out that the erodibility factor from Ginoux
et al. (2001) over the Gobi desert is significantly underestimated, which may result in the underestimation of the dust
emission over the Gobi desert. Given that simulated dust emission flux is directly scaled by erodibility factor, we suggest
that the erodibility factor used in our model needs to be updated or improved.
As stated above, although the model can capture the overall temporal variations of surface dust concentrations, the
occurrence of modeled peak values show earlier (about six hours) than the observations over several stations, which may
be attributed to strong transport due to stronger surface wind used in the model. It should be noted here that this model bias
contributes a lot to the simulation error, leading to smaller R and greater MB, NMB values. R values will be greatly
improved if this bias is eliminated, implying that the input assimilated meteorological field is important for dust emission
simulation and needs to be further evaluated and adjusted.
The averaged modeled surface $PM_{10}$ concentrations with and without the modifications (Sen_all and Control run
respectively) and observational values at ~1400 stations over China during the study period are compared in Fig. 9. It
shows that dust concentrations are generally underestimated in Control run (NMB=-16%, regression slope=0.4), which
could be attributed to the crude representation of soil properties, roughness length and other related elements. Incorporating
improvements in scheme makes the modeling result much closer to the observations, with R values increasing from 0.6 to
0.7, NMB values changing from -16% to -11%, regression slope ranging from 0.4 to 0.6. However, the improved model
still tends to underestimate the dust concentrations. Unrealistic soil properties (e.g., soil texture; roughness length) and



insufficiently accurate potential source map (the erodibility factor) used to scale dust emission flux could be the possible
causes.

To further investigate the model performance, spatial distributions of averaged simulated surface $PM_{10}$ concentrations
from Control run and Sen_all and their comparisons against observations are presented in Fig. 10. Results show that both
Control run and Sen_all can reproduce the pattern of dust concentrations in the study region, with high values located over
northwest China, North China and some areas of northeast China, indicating that GEOS-Chem can represent the main
features of dust emission and transport during the dust storm. It is found that for most sites in Control run, the simulated
magnitude are close to the observational values, but are underestimated over northwest China (where Gobi and Taklimakan
deserts are located) and North China plain. The simulated values from Sen_all are generally larger than Control run, and
are    more    consistent    with    measurements    both    in    magnitude    and    in    area    extent,    especially
over the desert region of northwest China. However, dust concentrations are still underestimated over North China plain,
possibly due to outdated source map or some potential dust source regions over Inner Mongolia are not well included. In
addition, missing mechanism of secondary aerosol source in the model such as heterogeneous reactions could also cause
the model bias (Zheng et al., 2015; Cheng et al., 2016).

Fig. 11 shows the spatial distribution of simulated averaged AOD from Control run and Sen_all as well as MODIS
AOD for the study period. For better comparison, simulated AOD at 13:00 local time (Aqua passage time) are extracted.
Result shows that Control run reproduces the major regions with high AOD values, e.g., eastern China, but with lower
magnitude. Control run also fails to capture the high-AOD area over the Taklimakan desert, while Sen_all could capture it.
Compared with Control run, Sen_all generally reproduce the spatial coverage and magnitude of the observed AOD.
**3.3 Discussion**

In our study, we point out that the erodibility factor (S) in the model may introduce uncertainty in modeled dust
concentrations, especially over Gobi desert. Several studies indicated that S from Ginoux et al.(2001) over the Gobi Desert
has been significantly underestimated and needs to be improved (Su and Fung, 2015; Zeng et al., 2020). Wu and Lin (2014)
have demonstrated that the potential source regions in the southeast of Mongolia and the middle-east of Inner Mongolia
are not well characterized by the S from GOCART scheme, which results in the underestimation of dust concentration in
this area and its downwind regions. In addition, the source function may not provide precise enough information about the
recent expansion of dust source areas over northern China, with the desertification and deforestation (Ku and Park, 2013).
Studies have demonstrated that implementing a physically based parameterization instead of an empirical dust source
function which is usually time-invariant and lacks physical treatment (Kok et al., 2014a, 2014b), or adopting the dynamic
dust source function (Xi et al., 2015), could improve the representation of dust emission. Therefore, the dust source function
should be precisely established with new updates and higher resolution using various measurements.

In terms of sandblasting efficiency α, many modeling studies as well as observational analysis have investigated its
magnitude and expression, but the results may vary greatly (Kang et al., 2011; Ma et al., 2019). The formulation for α used
in our improved scheme is based on LS99, which establishes the relationship between α and $u_*$, along with other soil-
related parameters dependent on soil textures. In this study, we derived α for different soil types based on the reported
values, but uncertainties still remain due to limited available measurements. In addition to the expression from MB95 and
LS99, there are other α formulations proposed by Shao et al. (1996) (Shao96) and Shao04. Different from the empirical
function, expressions of Shao96 and Shao04 are more sophisticated, which is the function of $u_{*t}$ and $u_*$ respectively,
along with some size information of soil particles. Comparisons of different formulations of α for different soil types (Kang
et al., 2011; Ma et al., 2019) have shown that the variation in α can reach up to several orders of magnitude, and few
equations could reproduce the measured positive correlation between α and $u_*$, suggesting that α for different soil texture
should be further investigated and observed to improve the model accuracy.

In this study, surface conditions including erodibility factor, soil texture, clay content and surface roughness length



play a significant role in improving the model performance of $u_*$, $u_{*t}$ and $F$. We conclude that substituting globally fixed
values of these properties with more realistic and physical-based ones could reduce the model uncertainty and improve the
understanding of dust emission mechanism. In physically-based scheme, the importance of accurate input surface
properties, including soil particle size distribution (Darmenova et al., 2009; Kok, 2011a, 2011b), soil texture (Shao et al.,
2011; Foroutan et al., 2017), surface roughness length (Darmenova et al., 2009; Kontos et al., 2018) and soil plastic pressure
(Lu and Shao, 1999; Kang et al., 2011), have also been highlighted by many studies. Therefore, accurate and abundant
observation data of soil-related properties are urgently needed, particularly over dust source region. Moreover, various and
comprehensive observation methods (e.g., experimental data, field and satellite observations) are recommended in order
to correct and update the input data.

## 4 Summary and Conclusion

In this study, we revised the treatments of dust emission processes by considering the effect of soil moisture, surface
roughness, soil texture, as well as Owen effect and more physically-based formulation of sandblasting efficiency in GEOS-
Chem version 12.6.0, in order to improve dust simulation over China. Several sensitivity simulations were conducted
during a severe dust storm between March 27, 2015 to April 2, 2015 over northern China to analyze the effects of these
modifications on $u_*$, $u_{*t}$ and emission flux.
In the improved scheme, we substituted global constant value of $Z_0$, assumed $M_{clay}$ in the original version with
geographical variation map obtained from the measurement provided by Prigent et al. (2005) and Shangguan et al. (2014)
respectively. $Z_{0s}$ and sandblasting efficiency were calculated with formulations based on soil texture data from FAO dataset,
which is more physically-based than the original version. In addition, Owen effect and drag partition correction factor were
considered in the improved version.
Sensitivity result showed that the modified $f_d$ and $f_w$ by inclusion of the updated $Z_0$, $Z_{0s}$ and $M_{clay}$ resulted in
significant alteration in $u_{*t}$ (ranging from -8%~72%) over China. $u_{*t}$ was increased when including the drag partition
effect, particularly with the updated $Z_0$ and $Z_{0s}$ field (increased by 10%~60%), which induced the modeled $u_{*t}$ much
closer to the measurements. Considering Owen effect increased modeled $u_*$ by 0%~39%, especially over northwest China
where surface wind is strong. In general, modeled $u_*$ was increased by 5%~50% over most parts of China due to the
inclusion of Owen effect and updated $Z_0$. In terms of sandblasting efficiency, it was decreased by 50% on average with
the updated $u_*$-dependent-ratio following LS99, with the largest reduction occurring over regions with sand texture. Due
to the combined effect of updated treatments, emission flux simulated by improved scheme was generally decreased over
northern China, except in some regions of northwest China, where Owen effect played a dominant role. Better agreement
between the improved model results and observational values was achieved in terms of the $u_{*t}$ and surface $PM_{10}$
concentrations in selected typical sites over northern China.
Compared with both surface $PM_{10}$ observations and MODIS AOD, the revised dust emission scheme produced better
performance in both temporal and spatial variation. Result showed that the dust concentrations were generally
underestimated at selected sites in the original scheme, particularly when dust concentrations were high. For the improved
scheme, both the magnitude and the temporal evolution of $PM_{10}$ concentrations were well captured, with peak values much
closer to the observations. According to the statistics, with the implementation of the updates, averaged $PM_{10}$ values at
selected sites were more comparable to the observations, and the average MB and NMB values were reduced from -
196.29µg $m^{-3}$ and -52.79% in Control run to -47.72µg $m^{-3}$ and -22.46% respectively. However, for some sites closer to
Gobi desert, dust concentrations were still underestimated, which was likely attributed to the uncertainty in the erodibility
factor over Gobi desert. Comparison of the model results and observed averaged $PM_{10}$ concentrations at ~1000 stations
showed that the revised scheme improved the model performance, with R values increasing from 0.6 to 0.7, NMB values
changing from -16% to -11%. Moreover, the improved scheme demonstrated better performance in reproducing the spatial
distribution of AOD than the original scheme, particularly over the desert region of northwest China.



In summary, this study indicated that compared to the original scheme, the revised dust emission scheme had an
overall better agreement with the measurements. However, more physically-based schemes and more detailed up-to-date
input parameters should be further investigated and observed to improve the accuracy of model.
*Code and data availability:* Measurements used in this work have been listed in Sect. 2.4 and acknowledgements. For
the model outputs and codes can be accessed by contacting Rong Tian (rongtian@nuist.edu.cn).
*Competing interests.* The authors declare that they have no conflict of interest.
*Author contributions.* RT designed and conducted the model experiments, analyzed the result and wrote the paper. XYM
supervised the project, proposed scientific suggestions and revised the paper. JQZ processed the observation data.
*Acknowledgements.* This study is supported by the National Natural Science Foundation of China grants (41675004 &
41975002), the National Key R&D Program of China grants (2019YFA0606802 & 2016YFA0600404), and the
Postgraduate Research & Practice Innovation Program of Jiangsu Province (grant no. SJKY19_0962). We are thankful to
C. Prigent for kindly providing the input map of global surface aerodynamic roughness length $Z_0$; to Shangguan W. for
providing the soil clay content data, which is available from the website of Land-Atmospheric Interaction Research Group
at Beijing Normal University (http://globalchange.bnu.edu.cn/research/soilw#download); to Sujan Koirala for providing
global soil texture map which can be downloaded at the website (http://hydro.iis.u-tokyo.ac.jp/~sujan/research/gswp3/soil-
texture-map.html). We are also grateful to GEOS-Chem Support Team for their management and maintenance of GEOS-
Chem model. We acknowledge to NASA, Chinese Ministry of Environmental Protection and Chinese National
Meteorological Center for providing the MODIS datasets, surface $PM_{10}$ observations and meteorological measurements
respectively.

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





Table 1. Input soil aggregate size distribution parameters dependent on soil texture classification following USDA.

| Soil Texture | Mode1 | | | Mode2 | | | Mode3 | | | $Z_{0s}$ |
| --- | --- | --- | --- | --- | --- | --- | --- | --- | --- | --- |
| | n | MMD | σ | n | MMD | σ | n | MMD | σ | |
| Sand | 0.9 | 1000 | 1.6 | 0.1 | 100 | 1.7 | 0 | 10 | 1.8 | 33.3 |
| Loamy sand | 0.6 | 690 | 1.6 | 0.3 | 100 | 1.7 | 0.1 | 10 | 1.8 | 23 |
| Sandy loam | 0.6 | 520 | 1.6 | 0.3 | 100 | 1.7 | 0.1 | 5 | 1.8 | 17.3 |
| Silt loam | 0.5 | 520 | 1.6 | 0.35 | 100 | 1.7 | 0.15 | 5 | 1.8 | 17.3 |
| Loam | 0.35 | 520 | 1.6 | 0.5 | 75 | 1.7 | 0.15 | 2.5 | 1.8 | 17.3 |
| Sandy clay loam | 0.3 | 210 | 1.7 | 0.5 | 75 | 1.7 | 0.2 | 2.5 | 1.8 | 7 |
| Silt clay loam | 0.3 | 210 | 1.7 | 0.5 | 50 | 1.7 | 0.2 | 2.5 | 1.8 | 7 |
| Clay loam | 0.2 | 125 | 1.7 | 0.5 | 50 | 1.7 | 0.3 | 1 | 1.8 | 4.2 |
| Sandy clay | 0.65 | 100 | 1.8 | 0 | 10 | 1.8 | 0.35 | 1 | 1.8 | 3.3 |
| Silty clay | 0.6 | 100 | 1.8 | 0 | 10 | 1.8 | 0.4 | 0.5 | 1.8 | 3.3 |
| Clay | 0.5 | 100 | 1.8 | 0 | 10 | 1.8 | 0.5 | 0.5 | 1.8 | 3.3 |
| Silt | 0.45 | 520 | 1.6 | 0.4 | 75 | 1.7 | 0.15 | 2.5 | 1.8 | 17.3 |

Including three-mode log-normal parameters (mass fraction n (%), mass median diameter MMD (μm), and geometric standard deviation
σ), and smooth aeolian roughness length $z_{0s}$ (μm).
Table 2. Input soil-related parameters for different soil texture used in calculation of sandblasting efficiency $\alpha$.

| Soil Texture | $p(N\ m^{-2})$ | $f(\%)$ | $\rho_b(\text{kg}\ m^{-3})$ | $C_\alpha$ |
| --- | --- | --- | --- | --- |
| Sand | 5000 | 6.9 | 1000 | 0.01 |
| Loamy sand | 5000 | 18.5 | 1000 | 0.008 |
| Sandy loam | 10000 | 22.3 | 800 | 0.7 |
| Silt loam | 10000 | 22.3 | 800 | 0.7 |
| Loam | 10000 | 22.3 | 800 | 0.7 |
| Sandy clay loam | 10000 | 22.3 | 800 | 0.9 |
| Silt clay loam | 10000 | 22.3 | 800 | 0.7 |
| Clay loam | 10000 | 22.3 | 800 | 0.9 |
| Sandy clay | 30000 | 72 | 700 | 0.2 |
| Silty clay | 30000 | 72 | 700 | 0.2 |
| Clay | 30000 | 72 | 700 | 0.2 |
| Silt | 10000 | 22.3 | 800 | 0.9 |






Table 3. Sensitivity Experiments design and description.

| Experiment name | Modifications | | | | | Description |
|---|---|---|---|---|---|---|
| | Updated $M_{clay}$ | Owen effect | Updated α | Drag partition correction (Default $Z_0$, $Z_{0s}$) | Updated $Z_0$, $Z_{0s}$ | |
| Control | N | N | N | N | N | Original scheme with default configurations. Serves as a control simulation. |
| Sen_mclay | Y | N | N | N | N | Adopting global $M_{clay}$ from Shangguan et al. (2014). |
| Sen_owen | N | Y | N | N | N | Considering Owen effect. |
| Sen_ratio | N | N | Y | N | N | Using updated α from Lu and Shao (1999). |
| Sen_drag | N | N | N | Y | N | Considering $f_d$ but with $Z_0$=0.01 cm, $Z_{0s}$=0.0033 cm |
| Sen_ $Z_0Z_{0s}$ | N | N | N | N | Y | Using updated $Z_0$ from Prigent et al. (2005) and updated $Z_{0s}$. |
| Sen_all | Y | Y | Y | Y | Y | Improved scheme including all the modifications described above. |


Table 4. Statistics for observed and simulated (Control and Sen_all) surface $PM_{10}$ concentrations at selected sites.

| Sites | Obs mean ($\mu g\ m^{-3}$) | Mod mean ($\mu g\ m^{-3}$) | | R | | MB ($\mu g\ m^{-3}$) | | NMB (%) | |
|---|---|---|---|---|---|---|---|---|---|
| | | Control | Sen_all | Control | Sen_all | Control | Sen_all | Control | Sen_all |
| Beijing | 232.33 | 130.54 | 148.90 | 0.17 | 0.15 | -87.40 | -64.78 | -37.62 | -27.88 |
| Tianjin | 196.68 | 121.86 | 135.89 | 0.01 | 0.02 | -72.87 | -52.46 | -37.05 | -26.67 |
| Huhehaote | 148.35 | 108.76 | 119.88 | 0.67 | 0.66 | -39.02 | -27.49 | -26.30 | -18.53 |
| Xilinguole | 116.51 | 48.35 | 64.11 | 0.56 | 0.57 | -72.47 | -73.17 | -62.20 | -62.80 |
| Kuele | 487.96 | 163.88 | 559.67 | 0.57 | 0.55 | -315.26 | 123.77 | -64.61 | 25.37 |
| Hami | 238.74 | 146.58 | 453.88 | 0.64 | 0.81 | -163.50 | -50.57 | -68.48 | -21.18 |
| Akesu | 738.39 | 236.09 | 827.03 | 0.45 | 0.50 | -503.61 | 79.95 | -68.20 | 10.83 |
| Jiuquan | 653.77 | 320.55 | 464.01 | 0.19 | 0.34 | -338.29 | -227.17 | -51.74 | -34.75 |
| Baiyin | 295.84 | 120.45 | 155.93 | 0.46 | 0.69 | -174.16 | -137.55 | -58.87 | -46.50 |
| Average | 345.40 | 155.23 | 325.48 | 0.41 | 0.48 | -196.29 | -47.72 | -52.79 | -22.46 |





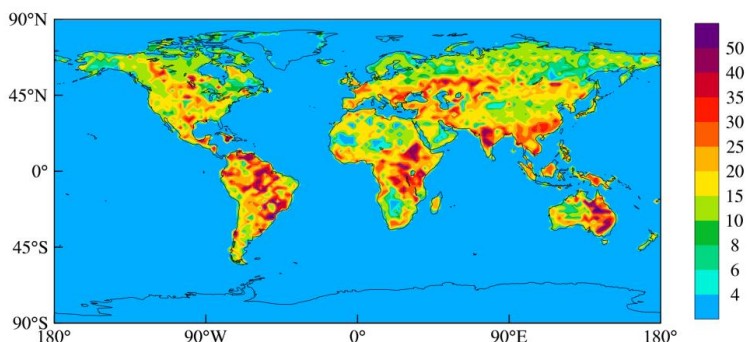

Fig 1. Updated input data of global $M_{clay}$ (%). Data is derived from Shangguan et al. (2014) and is re-gridded to $2° \times$

2.5° horizontal resolution in the model.


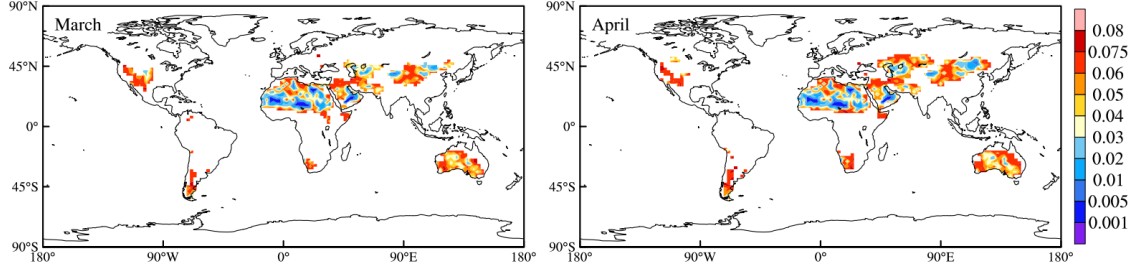

Fig 2. Monthly updated input data of global aerodynamic roughness length ($Z_0$) (cm) in March (left) and April (right).

Data is derived from Prigent et al. (2005) and re-gridded to $2° \times 2.5°$ horizontal resolution in the model.


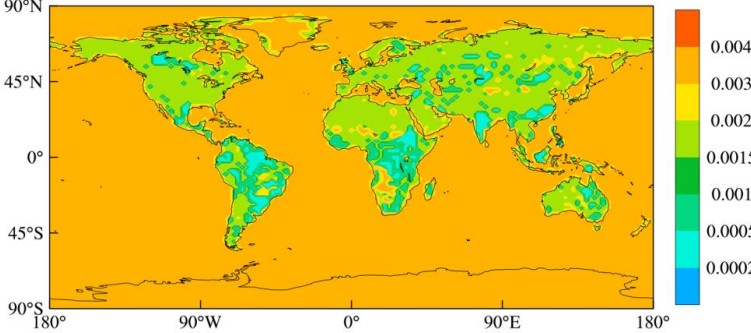

Fig 3. Updated global map of smooth roughness length ($Z_{0s}$) estimated from the empirical relationship with soil

texture.



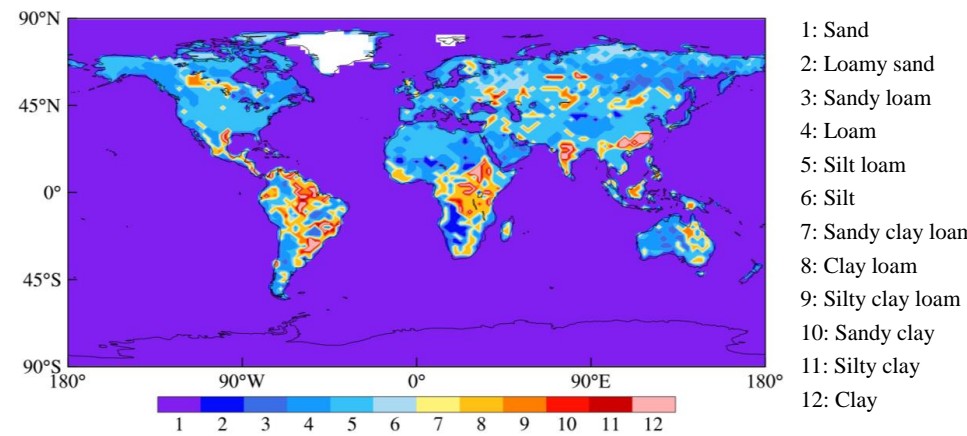

1: Sand

2: Loamy sand

3: Sandy loam

4: Loam

5: Silt loam

6: Silt

7: Sandy clay loam

8: Clay loam

9: Silty clay loam

10: Sandy clay

11: Silty clay

12: Clay


Fig 4. Global soil texture map based on the USDA classification.

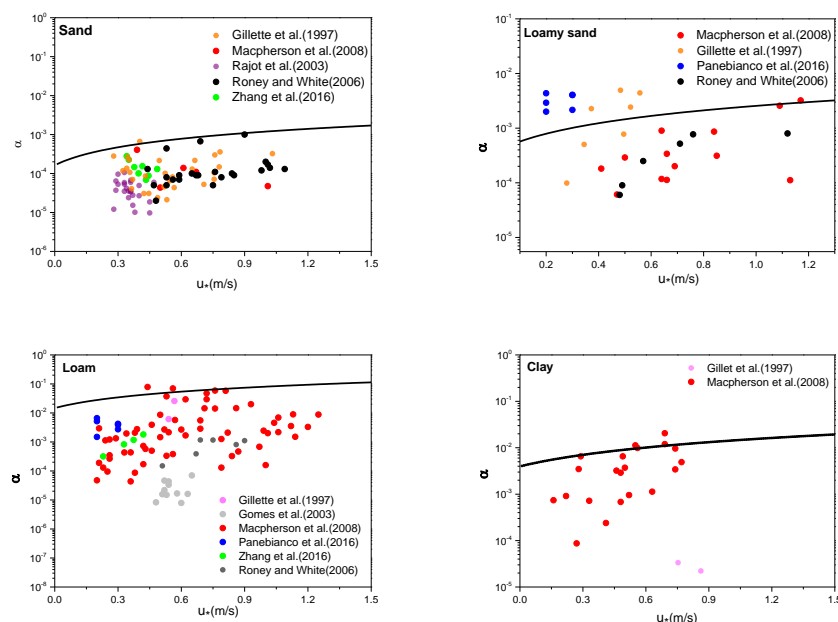


Fig 5. Updated sandblasting efficiency $\alpha$ as a function of surface friction velocity $u_*$ following Lu and Shao (1999)
for sand, loamy loam, loam and clay and observations from the literature.
Fig 6. Relative difference (%) in simulated averaged threshold friction velocity $u_{*t}$ (a), surface friction velocity $u_*$





(b), sandblasting efficiency $\alpha$ (c) and emission flux (d) between sensitivity simulations and control run during the
study period.

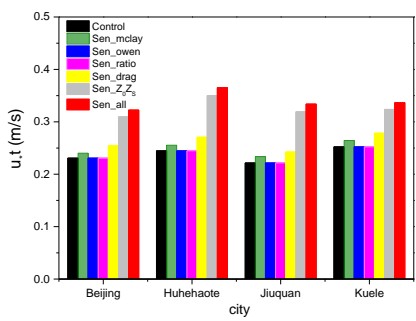 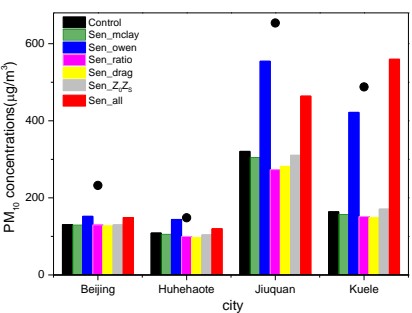

Fig 7. Comparisons of simulated averaged threshold friction velocity $u_{*t}$ (left) and PM$_{10}$ concentrations (right) at
selected sites. Black dots in right figure indicate the observed averaged PM$_{10}$ concentrations.

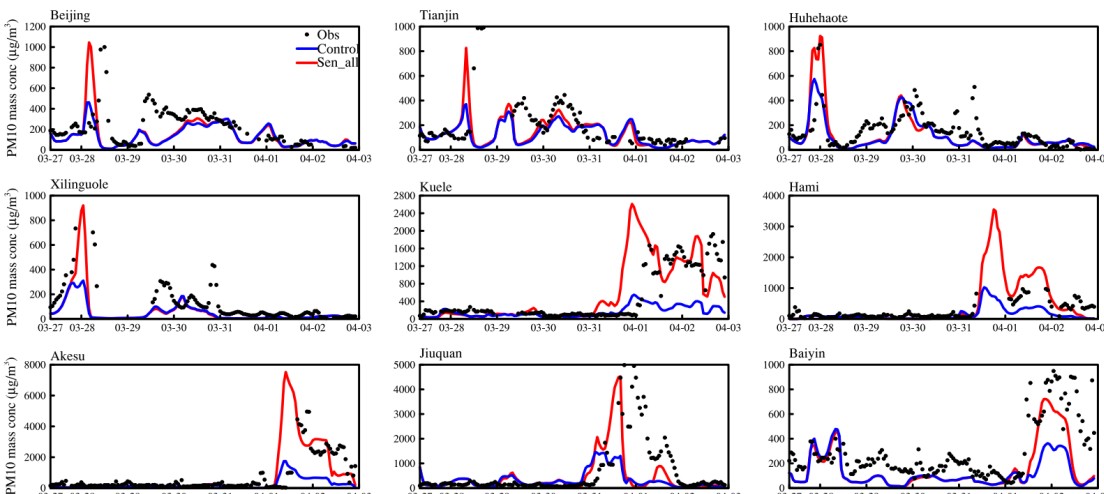

Fig 8. Temporal variation of hourly PM$_{10}$ concentrations from observations (black dots) and simulations of Control
run (blue line) and Sen_all (red line) during the study period at nine selected sites.



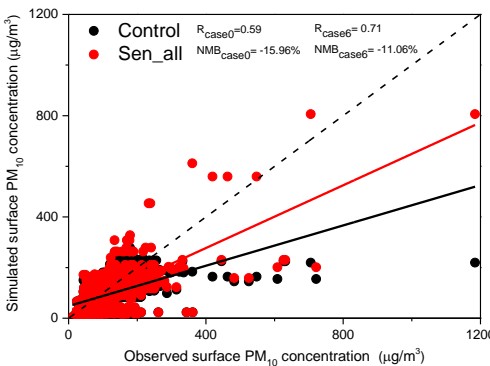

Fig 9. Comparison of modelled and measured surface $PM_{10}$ concentrations at observational sites. The dotted line is
the 1:1 line. Model results are taken from Control run and Sen_all respectively.

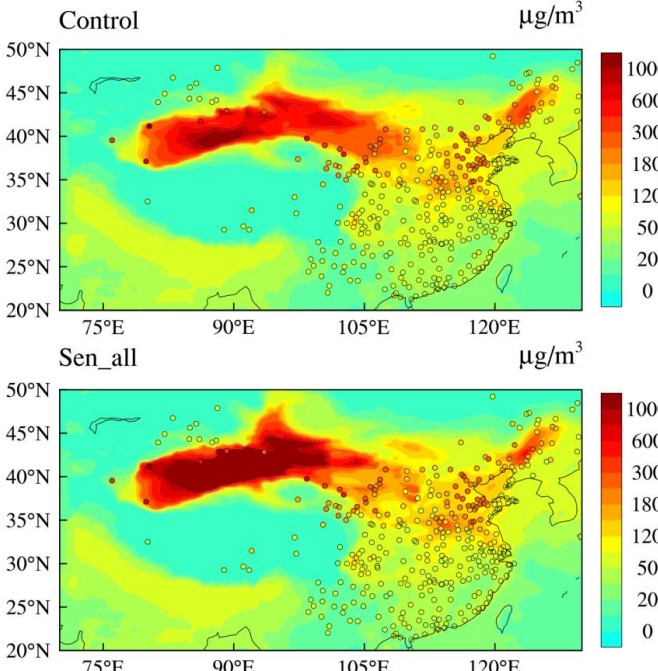


Fig 10. Comparison of simulated averaged $PM_{10}$ surface concentrations from Control run (top) and Sen_all (bottom)
with the observed values.






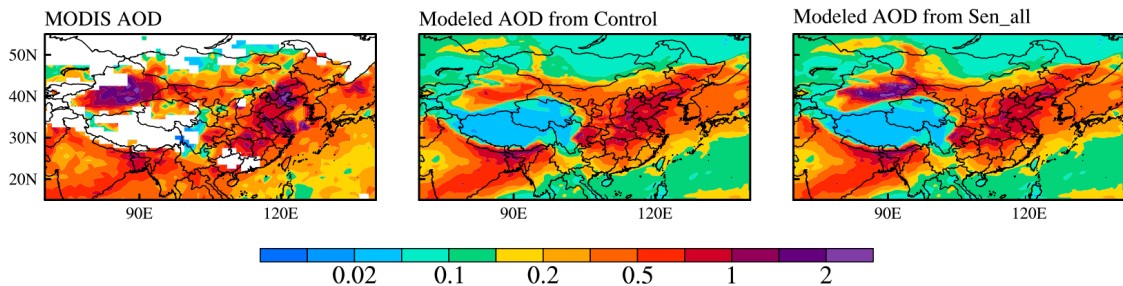


Fig 11. Spatial distribution of MODIS retrieved AOD at 550nm (left column) and simulated AOD at 550 nm from
Control run (middle column) and Sen_all (right column). The simulation results are extracted at 13:00 local time to
match the MODIS observation time.