# Peer review of "improvements in dust simulations over China"

_Atmospheric Chemistry and Physics, 2020_

## Referee Comment (RC1) · Anonymous Referee #1 · 30 Nov 2020

This study presents an improvement of the dust emission scheme in GEOS-Chem model by incorporating the updated soil texture and aerodynamic roughness length with spatial variability, Owen effect, drag partition correction factor as well as the updated formulation of sandblasting efficiency. Detailed model-observation comparisons are made in China. I think this paper is clearly written and organized and should be accepted in ACP.

---

## Referee Comment (RC2) · Anonymous Referee #2 · 7 Jan 2021

General Comments: This paper presents the simulations of the Asian dust with GEOS-Chem model. The main drawbacks of the original parameterization of the dust emission used in official GEOS-Chem are pointed out firstly, subsequently the authors make a lot of efforts to improve the dust emission scheme by revising parameters such as aerodynamic roughness length, soil texture, and sandblasting efficiency. The simulated spatial and temporal variations of dust aerosols are found much closer to observations with the revised GEOS-chem model. General speaking, the manuscript is scientifically sound and well organized. I recommend accepting it after addressing the following comments. Major comments: 1) I suppose you are using a nested version of GOES-Chem with higher model resolution over your target region East Asia. Are there any interactions between the global simulation and the nested region? Please clarify this.

[Figure]

2) How the dust size distributions are considered after the bulk vertical emission flux calculated? Specific comments: 1. Fig S2 is better for reader to understand your study. I suggest you moving it to the main text. What is the meaning of the $u_{10,t}$ in Fig S2? 2. What is the meaning of the contour plot in Fig. S5? 3. The units of $Z_{0s}$ in Table 1 and Fig. 3 are inconsistent, please clarify the unit in Fig. 3. 4. In Fig.7, it is meaningless to compare the simulated averaged threshold friction velocities in Beijing, since there are no dust emissions in Beijing due to the erodibility factor S. Therefore, I recommend you making more comparisons over the dust source regions.

---

## Author Comment (AC2) · 26 Jan 2021

Thanks to the reviewers for their constructive comments and very helpful suggestions, which have allowed us to clarify and improve the manuscript. Below we address the reviewers' comments, with the reviewer comments in black, and our responses in blue. We have revised the manuscript accordingly.

**Reviewer #2:**

General Comments:

This paper presents the simulations of the Asian dust with GEOS-Chem model. The main drawbacks of the original parameterization of the dust emission used in official GEOS-Chem are pointed out firstly, subsequently the authors make a lot of efforts to improve the dust emission scheme by revising parameters such as aerodynamic roughness length, soil texture, and sandblasting efficiency. The simulated spatial and temporal variations of dust aerosols are found much closer to observations with the revised GEOS-Chem model. General speaking, the manuscript is scientifically sound and well organized. I recommend accepting it after addressing the following comments.

**Major comments:**

1) I suppose you are using a nested version of GOES-Chem with higher model resolution over your target region East Asia. Are there any interactions between the global simulation and the nested region? Please clarify this.

Thanks for the suggestion. In GEOS-Chem, running a nested simulation requires the first step of running a global simulation with a coarse resolution. The global simulation is conducted to generate boundary conditions which is used to initialize species concentrations at the boundaries of our nested grid region, but not vice versa (Wang et al., 2004). Therefore, it is a one-way nesting procedure (that is to say, the results from the global model is only used to define the boundary conditions for nested simulation, but the nested simulation has no feedback on the global simulation). Many nested GEOS-Chem simulations have been conducted over different regions, e.g., Asia (Li et al., 2013; Lin et al., 2014; Zhang et al., 2015; Dang and Liao, 2019), North America (Heald et al., 2012; Jiang et al., 2015; Fisher et al., 2016), and Europe (Tombrou et al., 2009; Vinken et al., 2014). Both gaseous and aerosol species have been simulated and evaluated by previous work (e.g., Wang et al., 2004; Chen et al., 2009; Jeong et al., 2011; Heald et al., 2012; Wang et al., 2013; Li et al., 2019), showing that nested version of GEOS-Chem exhibited good agreement with the measurements.

We have included the associated description in the revised manuscript (lines 93-99).

2) How the dust size distributions are considered after the bulk vertical emission flux calculated?

Mineral dust aerosols in GEOS-Chem are simulated across 4 size bins (radii 0.1–1.0, 1.0–1.8, 1.8–3.0, and 3.0–6.0 μm). We adopted the dust particle size distribution (PSD) proposed by Zhang et al. (2013) after the calculation of dust emission flux. As described by Zhang et al. (2013), mass fractions of each size bins

are 7.7%, 19.2%, 34.9% and 38.2% accordingly. This parameterization is recommended by GEOS-Chem Aerosols Working Group, and has been evaluated for dust over United States and Asia, etc. (Zhang et al., 2013; Philip et al., 2017; Yumimoto et al., 2017; Latimer et al., 2019).
We have included the associated description in the revised manuscript (lines 102-104).

**Specific comments:**

1) Fig S2 is better for reader to understand your study. I suggest you moving it to the main text. What is the meaning of the u10,t in Fig S2?
Thanks for suggestion. We have moved Fig S2 to the main text (Fig. 1 in the revised manuscript). $u_{10,t}$ in the figure represents the threshold saltation wind speed at 10m, which is calculated by wind speed at 10m ($u_{10m}$), surface friction velocity($u_*$) and threshold friction velocity($u_{*t}$):

$$u_{10,t} = \frac{u_{10m} \times u_{*t}}{u_*}$$

We have included the description in the figure.

2) What is the meaning of the contour plot in Fig S5?
Figure S5 displays the comparisons of averaged surface wind field between the model input and observations. It is used to show that the circulation patterns in the model are identical with the observations, with surface wind speed in the model larger than observations to some extent, which was also found by Wang et al. (2014). We have referred this figure in the manuscript (lines 227-230 in the revised manuscript).

3) The units of Z0s in Table 1 and Fig. 3 are inconsistent, please clarify the unit in Fig. 3.
Thanks for reminder. We have modified the unit of $Z_{0s}$ to cm in Table 1 to make the units in the full-text consistent.

4) In Fig.7, it is meaningless to compare the simulated averaged threshold friction velocities in Beijing, since there are no dust emissions in Beijing due to the erodibility factor S. Therefore, I recommend you making more comparisons over the dust source regions.
Thanks for suggestion. Yes, we agree. In the revised manuscript, we have removed the comparison of Beijing in this figure, and added the comparisons over Xilinguole and Akesu sites, which are located over the dust source regions (seen in lines 272-276 and Fig. 8 in the revised version).

**References**

Chen, D., Wang, Y., McElroy, M. B., He, K., Yantosca, R. M., and Le Sager, P.: Regional CO pollution and export in China simulated by the high-resolution nested-grid GEOS-Chem model, Atmos. Chem. Phys., 9, 3825-3839, 10.5194/acp-9-3825-2009, 2009.

Dang, R. and Liao, H.: Severe winter haze days in the Beijing–Tianjin–Hebei region from 1985 to 2017 and the roles of anthropogenic emissions and meteorology, Atmos. Chem. Phys., 19, 10801–10816, https://doi.org/10.5194/acp-19-10801-2019, 2019.

Fisher, J. A., Jacob, D. J., Travis, K. R., Kim, P. S., Marais, E. A., Miller, C. C., Yu, K., Zhu, L., Yantosca, R. M., and Sulprizio, M. P.: Organic nitrate chemistry and its implications for nitrogen budgets in an isoprene- and monoterpene-rich atmosphere: constraints from aircraft (SEAC4RS) and ground-based (SOAS) observations in the Southeast US, Atmos. Chem. Phys., 16, 1-38, 2016.

Heald, C. L., J. L. Collett, J., Lee, T., Benedict, K. B., Schwandner, F. M., Li, Y., Clarisse, L., Hurtmans, D. R., Van Damme, M., Clerbaux, C., Coheur, P. F., Philip, S., Martin, R. V., and Pye, H. O. T.: Atmospheric ammonia and particulate inorganic nitrogen over the United States, Atmos. Chem. Phys., 12, 10295-10312, 10.5194/acp-12-10295-2012, 2012.

Jeong, J. I., Park, R. J., Woo, J. H., Han, Y. J., and Yi, S. M.: Source contributions to carbonaceous aerosol concentrations in Korea, Atmos. Environ., 45, 1116-1125, 2011.

Jiang, Z., Jones, D. B. A., Worden, J., Worden, H. M., Henze, D. K., and Wang, Y. X.: Regional data assimilation of multi-spectral MOPITT observations of CO over North America, Atmos. Chem. Phys., 15, 5327-5358, 2015.

Latimer, R. N. C. and Martin, R. V.: Interpretation of measured aerosol mass scattering efficiency over North America using a chemical transport model, Atmos. Chem. Phys., 19, 2635–2653, https://doi.org/10.5194/acp-19-2635-2019, 2019.

Li, K., Jacob, D. J., Liao, H., Zhu, J., Shah, V., Shen, L., Bates, K. H., Zhang, Q. and Zhai, S.: A two-pollutant strategy for improving ozone and particulate air quality in China, Nat. Geosci., 12(11), 906–910, https://doi.org/10.1038/s41561-019-0464-x, 2019.

Li, M., Wang, Y., and Ju, W.: Effects of a Remotely Sensed Land Cover Dataset with High Spatial Resolution on the Simulation of Secondary Air Pollutants over China Using the Nested-grid GEOS-Chem Chemical Transport Model, Advances in Atmospheric Sciences, 31, 179-187, 2013.

Lin, J., Xin, J., Che, H., Wang, Y., and Donkelaar, A. V.: Clear-sky aerosol optical depth over East China estimated from visibility measurements and chemical transport modeling, Atmos. Environ., 95, 258-267, 2014.

Philip, S., Martin, R. V., Snider, G., Weagle, C. L., van Donkelaar, A., Brauer, M., Henze, D. K., Klimont, Z., Venkataraman, C., Guttikunda, S. K. and Zhang, Q.: Anthropogenic fugitive, combustion and industrial dust is a significant, underrepresented fine particulate matter source

in global atmospheric models, Environ. Res. Lett., 12(4), 044018, https://doi.org/10.1088/1748-9326/aa65a4, 2017.

Tombrou, M., Bossioli, E., Protonotariou, A. P., Flocas, H., Giannakopoulos, C., and Dandou, A.: Coupling GEOS-CHEM with a regional air pollution model for Greece, Atmos. Environ., 43, 4793-4804, 2009.

Vinken, G. C. M., Boersma, K. F., van Donkelaar, A., and Zhang, L.: Constraints on ship NOx emissions in Europe using GEOS-Chem and OMI satellite NO2 observations, Atmos. Chem. Phys., 14, 1353-1369, 10.5194/acp-14-1353-2014, 2014.

Wang, Y. X., Mcelroy, M. B., Jacob, D. J., and Yantosca, R. M.: A nested grid formulation for chemical transport over Asia: Applications to CO, J. Geophys. Res. Atmos., 109, D22307, 10.1029/2004JD005237, 2004.

Wang, Y., Zhang, Q. Q., He, K., Zhang, Q., and Chai, L.: Sulfate-nitrate-ammonium aerosols over China: response to 2000–2015 emission changes of sulfur dioxide, nitrogen oxides, and ammonia, Atmos. Chem. Phys., 13, 2635-2652, 10.5194/acp-13-2635-2013, 2013.

Wang, Y., Zhang, Q., Jiang, J., Zhou, W., Wang, B., He, K., Duan, F., Zhang, Q., Philip, S. and Xie, Y.: Enhanced sulfate formation during China's severe winter haze episode in January 2013 missing from current models: MODELING WINTER HAZE FORMATION IN CHINA, J. Geophys. Res. Atmos., 119(17), 10,425-10,440, https://doi.org/10.1002/2013JD021426, 2014.

Yumimoto, K., Uno, I., Pan, X., Nishizawa, T., Kim, S.-W. and Sugimoto, N.: Inverse Modeling of Asian Dust Emissions with POPC Observations: A TEMM Dust Sand Storm 2014 Case Study, SOLA, 13(0), 31–35, https://doi.org/10.2151/sola.2017-006, 2017.

Zhang, L., Kok, J. F., Henze, D. K., Li, Q. and Zhao, C.: Improving simulations of fine dust surface concentrations over the western United States by optimizing the particle size distribution: IMPROVING SIMULATED DUST OVER WESTERN US, Geophys. Res. Lett., 40(12), 3270–3275, https://doi.org/10.1002/grl.50591, 2013.

Zhang, L., Liu, L., Zhao, Y., Gong, S., Zhang, X., Henze, D. K., Capps, S. L., Fu, T.-M., Zhang, Q., and Wang, Y.: Source attribution of particulate matter pollution over North China with the adjoint method, Environmental Research Letters, 10, 084011, 10.1088/1748-9326/10/8/084011, 2015.

---

## Author Comment (AC1)

Dear Editor and Referees,

Thanks for giving us an opportunity to revise our manuscript entitled " A revised mineral dust emission scheme in GEOS-Chem: improvements in dust simulations over China "(ID: acp-2020-984). Thanks for editor's effort in handling the review of our manuscript.

We appreciate your positive and constructive comments. We have studied these comments carefully and make revisions on the manuscript. The point-to-point comments and corresponding responses are attached below. The manuscript has been revised accordingly.

Thank you again for your time and effort. We look forward to hearing from you soon.

Best regards,

Yours sincerely,

Rong Tian, Xiaoyan Ma, and Jianqi Zhao

REVIEWER COMMENTS

Thanks to the reviewers for their constructive comments and very helpful suggestions, which have allowed us to clarify and improve the manuscript. Below we address the reviewers' comments, with the reviewer comments in black, and our responses in blue. We have revised the manuscript accordingly.

**Reviewer #1:**

This study presents an improvement of the dust emission scheme in GEOS-Chem model by incorporating the updated soil texture and aerodynamic roughness length with spatial variability, Owen effect, drag partition correction factor as well as the updated formulation of sandblasting efficiency. Detailed model-observation comparisons are made in China. I think this paper is clearly written and organized and should be accepted in ACP.

Thanks to the reviewer for the comments.